# Chitosan-Based Semen Extenders: An Approach to Antibiotic-Free Artificial Insemination in Rabbit

**DOI:** 10.3390/antibiotics14010055

**Published:** 2025-01-09

**Authors:** Francisco Marco-Jiménez, Celia Ferriz-Nuñez, Maria Pilar Viudes-de-Castro, José Salvador Vicente, Laura Lorenzo-Rebenaque

**Affiliations:** 1Instituto de Ciencia y Tecnología Animal, Universitat Politècnica de València, C/Camino de Vera S/N, 46022 Valencia, Spain; cefernue@gmail.com (C.F.-N.); jvicent@dca.upv.es (J.S.V.); laulore@upv.es (L.L.-R.); 2Centro de Investigación y Tecnología Animal, Instituto Valenciano de Investigaciones Agrarias (CITA-IVIA), Polígono La Esperanza No. 100, 12400 Segorbe, Spain; viudes_mar@gva.es

**Keywords:** ethylenediaminetetraacetic acid, sperm function, *Enterococcus faecalis*, fertility, total kits born, embryonic losses, fetal losses

## Abstract

**Background/Objectives**: The use of antibiotics in livestock contributes to antimicrobial resistance, highlighting the need for alternative solutions. Among these, chelating agents, like ethylenediaminetetraacetic acid (EDTA) and Chitosan, have shown potential in reducing bacterial contamination in seminal doses used in artificial insemination (AI), while preserving sperm quality. The objective of this study was to evaluate the potential use of EDTA and Chitosan as alternatives to antibiotics for the liquid storage of rabbit seminal AI doses. **Methods**: EDTA (20 mM) and Chitosan (0.05%) were tested both individually and in combination, by adding them to the semen extender, and their effects were compared with extenders containing antibiotics or none. The extenders were evaluated for microbial resistance and their ability to maintain sperm quality in vitro during refrigeration at 16 ± 1 °C for 72 h. To assess antimicrobial efficacy, Enterococcus faecalis was used. Seminal doses stored for 24 h were used for insemination under commercial conditions, and fertility rate and total kits born were evaluated. **Results**: Adding 0.05% Chitosan to the extender resulted in sperm parameters and bacterial load comparable to those achieved with antibiotics during refrigerated storage, yielding similar fertility rate and total kits born outcomes 24 h post-storage. In contrast, the use EDTA alone or in combination with Chitosan was less effective at controlling *Enterococcus faecalis* than the antibiotic extenders, which also resulted in a reduction of sperm total motility over storage period (0–72 h) and negatively impacted fertility rate and total kits born. **Conclusions**: Chitosan’s protective effect on sperm function, combined with its antimicrobial activity, makes it a promising alternative antimicrobial agent for the liquid storage of rabbit seminal AI doses.

## 1. Introduction

The use of antibiotics in semen extenders has been a standard practice for many years, dating back to the onset of commercial artificial insemination (AI) in livestock several decades ago [1]. A target has been set by the European Union for a 50% reduction in overall sales of antimicrobials for farmed animals by 2030, as part of the Farm to Fork Strategy and the Zero Pollution Action Plan. The reduced use of antimicrobials in farmed animals should be monitored through the Common Agricultural Policy support measures [2]. Ideally, antimicrobials should only be prescribed for therapeutic purposes [3]. Therefore, the non-therapeutic use of antimicrobial agents in semen doses conflicts with current prudent-use recommendations advocated by medical and veterinary professionals. Antimicrobial resistance (AMR) has emerged as one of the most significant global health threats, increasing the risks of disease transmission, severe illness, and mortality. In bacteria, where AMR occurs naturally, the misuse and overuse of antibiotics accelerate this process, undermining the ability of modern medicine to treat infections [4]. Addressing AMR is inherently complex, as it involves intricate interactions among humans, animals, and the environment, consistent with the One Health approach [5].

As in most livestock species, rabbit breeding relies on the use of AI with diluted ejaculates, which involves the collection, processing, and preservation of male gametes [6]. It is increasingly recognized that the presence of microorganisms in semen is not solely due to infection during collection and processing but rather reflects the existence of a microbiome in each biofluid [7], with interactions occurring between host genetics and the seminal microbiome [8]. Semen extenders are essential to maintaining sperm survival until insemination, but their nutrient content also supports bacterial survival [9]. Consequently, antibiotics are routinely added to semen extenders, used to prepare seminal doses to prevent bacterial growth [10]. However, the lack of standardized antimicrobial prophylaxis guidelines for the preparation and distribution of seminal doses may also contribute to the global issue of AMR. The rational use of antibiotics in semen extenders and the replacement of conventional antibiotics with alternative strategies are two key approaches to managing unavoidable bacterial contamination, mitigating its detrimental effects on semen quality, and minimizing the global threats posed by antimicrobial resistance [4,11].

Possible alternatives to antibiotics in semen extenders were recently reviewed [1,12], where strategies such as temperature reduction, centrifugation in colloids [13,14,15] and the addition of antimicrobial substances were described. Unfortunately, neither semen freezing nor centrifugation is a feasible strategy in rabbits due to the adverse effects on spermatozoa. Although recent results demonstrate that rabbit semen freezing is feasible by combining Me2SO (10.7%) and dextran (5%) in terms of fertility rate and total kits born [16], its applicability in routine livestock practices is not highly viable due to the additional costs of the doses. Regarding centrifugation, the prostate granules present in seminal plasma prevent the sperm acrosome reaction. When the granules are removed through centrifugation with colloids, there is a significant decrease in fertilizing ability [17]. Based on this, the most feasible approach to exploring the possible elimination of antibiotics in semen doses is through the addition of antimicrobial substances [18]. In rabbits, the addition of chelating substances has shown promising results as an alternative to the use of antibiotics in seminal doses [19]. The inclusion of EDTA is a widely used method to inhibit calcium’s role as a mediator in sperm capacitation and acrosome reaction [20]. Therefore, semen extenders containing chelating agents, like EDTA, may serve as effective alternatives to antibiotics for semen preservation in the fight against AMR. Bestatin is an aminopeptidase produced by *Actinomycetes* spp. that exhibits antimicrobial properties [21]. However, antimicrobial substances may not be effective in all situations or may be spermatotoxic which is counter-productive [1]. Chitosan, a biocompatible biodegradable and non-toxic polycationic copolymer extensively used as material for encapsulation and controlled release of chemicals [22], interacts with the bacterial cell membrane and causes cell lysis [23,24].

The main objective of this study was to assess the potential use of two chelating agents, EDTA and Chitosan, added to semen extenders as possible alternative to antibiotics for the liquid storage of rabbit seminal AI doses, in alignment with the standards of rabbit semen production centers.

## 2. Results

### 2.1. Antimicrobial Extender Activity

Bacterial growth of *Enterococcus faecalis* is shown in Figure 1. Susceptibility was observed only for EDTA and +AB extenders. The extender supplemented with Chitosan, as well as the extender containing EDTA and Chitosan and no antibiotics, did not appear to have the ability to inhibit growth both at both 1 × 10^3^ and 10^5^ CFU/mL cultures at 37 °C during 24 h.

### 2.2. Bacterial Counts Determination

As shown in Figure 2, the Chitosan values of bacterial cell cultures are similar to the +AB extender. In contrast, neither the extender supplemented with EDTA nor the extender containing both substances appears to have the ability to inhibit growth. Specifically, although at 24 h of refrigeration they showed similar growth to the group without antibiotics (*p* > 0.05), this ability is lost at both 48 and 72 h. In the specific case of the extender with EDTA, growth was even higher than that of the extender without antibiotics (*p* < 0.05).

### 2.3. Experiment 1: In Vitro Evaluation

Figure 3A shows sperm motility across extenders over a 72-h storage period. Total sperm motility in the presence of EDTA (EDTA and EDTA and Chitosan) was significantly lower (*p* < 0.05) than in all other extenders from 0 to 72 h. In contrast, the percentage of motile spermatozoa in the Chitosan extender was comparable to that in the antibiotic-supplemented control. Regarding viability, no differences were observed between the different extenders over the 72 h of storage (Figure 3B).

### 2.4. Experiment 2: In Vivo Evaluation

Reproductive outcomes are shown in Table 1.

The pregnancy rate and the total kits born were influenced by the extender used. While EDTA resulted in the lowest fertility rate compared to +AB, EDTA and Chitosan showed the lowest prolificacy compared to +AB (Table 2). However, the addition of Chitosan produced results similar to those observed with the addition of antibiotics.

The effects of the extenders on embryonic and fetal losses indicate that EDTA and Chitosan led to the highest rates of embryonic and fetal losses compared to +AB (Table 3). The other extenders produced results comparable to those observed with the use of antibiotics.

## 3. Discussion

Our findings provide strong evidence that adding 0.05% Chitosan to the TCG extender results in sperm parameters and bacterial load comparable to those achieved with penicillin (100 IU/mL) and streptomycin (100 µg/mL) during the refrigerated storage of semen doses, yielding similar pregnancy rate and prolificacy (total kits born) results when AI was performed at 24 h of storage. In contrast, EDTA at 20 mM and the combination of EDTA and Chitosan in the TCG extender were less effective in controlling *Enterococcus faecalis* compared to extenders with antibiotics. Additionally, these treatments significantly reduced sperm motility percentages over 72 h of storage compared to the antibiotics group. This reduction was accompanied by a marked negative impact on pregnancy rate at day 12 (EDTA) and prolificacy, attributed to the high rates of embryonic and fetal loss observed with EDTA and Chitosan. The absence of toxicity of Chitosan to sperm cells and the sperm fertilizing ability, make this chelating agent a potential alternative to the antibiotics present in commercial seminal doses in rabbit AI industry.

When studying the different extenders for bacterial growth test at 37 °C over 24 h, it can be concluded that only EDTA was capable of completely inhibiting the growth of *Enterococcus faecalis* regardless of the concentration, similar to antibiotics. However, when the contaminated extenders were stored under refrigeration conditions (16 ± 1 °C for 72 h), both the EDTA extender and the combination of EDTA and Chitosan failed to prevent *Enterococcus faecalis* proliferation over the 72-h period and also reduced sperm motility compared to the antibiotic extender. Moreover, Chitosan showed no significant differences in bacterial load (at 24, 48, and 72 h) or sperm motility percentages when compared to the antibiotic extender. Chelating agents as alternative antibiotics offer a promising strategy for semen preservation, as they are less likely to contribute to bacterial resistance due to their unique mechanism of action [20,25,26]. This mechanism is similar to that of nitroxoline, one of the oldest antibiotics, which does not belong to any conventional antibacterial class [27,28]. The term “chelate”, derived from a Greek word for crab claw, refers to a group of complex (or coordination) compounds containing polydentate ligands. Polydentate ligands, from the Greek roots *poly* (meaning “many”), *dent* (meaning “teeth”), and *ligare* (meaning “to bind”), are molecules or ions that surround a central metal cation, attaching to it with at least two coordinate covalent bonds, thereby forming a chelate [29]. The connection between chelation and antimicrobial activity, as seen in nitroxoline’s mode of action, is rooted in the fact that (transition) metal cations are vital for bacterial survival [30]. EDTA is a well-known chelating agent, biocompatible at pH 7, whose main characteristic is its ability to chelate metallic ions essential for microbial growth, leading to microbial death, despite having no direct antibacterial effect [31]. However, EDTA has a strong calcium-chelating ability with four carboxylic acid groups [32]. The inclusion of EDTA in semen extenders is a common practice to block calcium’s role as a mediator in sperm capacitation and the acrosome reaction [20]. Additionally, it could be leveraged to broaden the antimicrobial spectrum of antimicrobial peptide preparations AMPPs [18]. Therefore, semen extenders containing chelating agents, like EDTA, may offer a viable alternative in the fight against AMR by replacing traditional antibiotics in semen preservation [18]. EDTA in combination with bestatin has been previously tested in rabbit semen showing a significant reduction against *Enterobacteriaceae* (Gram-negative) without toxicity to the sperm cells [19,33].

In our study, we did not observe such phenomena, as we even found a significant decline in motility, fertility rate and total kits born compared to the extender with antibiotics. These differences could be attributed to the use of *Enterococcus faecalis* (Gram-positive) as a sentinel in this study or to the combination with bestatin. Bestatin, an antibiotic of microbial origin, is a potent peptidase inhibitor that permeates cell membranes more readily and is useful for inhibiting intracellular enzymes [34]. An important point in this study was that AI was carried out 24 h after storage, whereas in all previous rabbit studies, insemination was performed immediately after the preparation of the semen doses [19,33,35]. These two scenarios are typical of rabbit livestock farmers who have bucks on their facilities and prepare their own seminal doses for immediate use. In contrast, when seminal doses are elaborate and from buck semen of the best genetics, the logistics involved in delivering them to the application farm require 24 h of storage. Another explanation is supported by the concentration added: while the addition of low concentrations of EDTA does not appear to produce negative effects—such as 3.36 mM in the porcine TBS diluent [36]—high concentrations (17.1 mM) act as a spermicidal agent due to their ability to modulate the calcium ion concentration in semen [37]. Indeed, its effect is evident immediately upon contact with sperm, demonstrating a high affinity for divalent cations such as Ca^2^⁺, which likely disrupts the delicate ionic balance within semen [37]. By chelating calcium ions and potentially altering the concentrations of sodium and potassium, EDTA interferes with ion homeostasis. This disruption can inhibit the Ca^2^⁺/Na⁺ exchanger on the sperm membrane, a critical mechanism for regulating calcium influx essential for sperm motility [37].

By contrast, Chitosan sustains *Enterococcus faecalis* growth and preserves sperm quality during semen storage, similar to the effects of antibiotics. Chitosan is a deacetylated product of chitin and attracts interest due to its easy availability, economic feasibility, ecofriendly, non-toxicity and biodegradability [38]. The precise antimicrobial action mechanism of Chitosan against Gram-positive and Gram-negative bacteria is not yet fully understood [39]. Chitosan exhibits antimicrobial activity through several mechanisms, including disruption of microbial membrane fluidity via electrostatic interactions, interference with transcription and translation by inhibiting mRNA and protein synthesis, chelation of metal ions to inhibit microbial growth, and prevention of microbial growth by blocking nutrient and gas exchange, with variations in activity potentially due to differences in microbial cell composition [39]. In addition to its antibacterial properties, Chitosan has demonstrated antifungal activity against certain fungal species. Research suggest that Chitosan can penetrate fungal hyphae and inhibit the key enzymes required for their growth and development [40]. In our study, we did not observe that Chitosan had the ability to inhibit growth at 37 °C; however, this ability was observed at 16 ± 1 °C from 0 to 72 h. These differences could be attributed to temperature. Researchers have investigated this and found that Chitosan stored at 4 °C for 15 weeks exhibited the strongest antimicrobial activity against several microbial strains compared to Chitosan stored at 25 °C [41]. The use of Chitosan does not affect semen quality and the capacitation status of spermatozoa in rabbit semen doses [42]. Chitosan is well-known for its antioxidant activity [43,44,45,46,47,48]. In recent years, there has been increasing interest in studying the effects of Chitosan and its derivatives on sperm plasma membrane lipids and lipid peroxidation, highlighting its potential role in protecting sperm from oxidative damage [49].

## 4. Materials and Methods

### 4.1. Extenders Composition

Five extenders were freshly prepared in tris-hydroxymethylaminometano (250 mM), citric acid (83 mM) and glucose (50 mM) (pH 6.8–7.0, TCG extender; [50]): EDTA (20 mM), Chitosan (0.05%), EDTA and Chitosan (20 mM and 0.05%), +AB (100 UI/mL of penicillin and 100 µg/mL of streptomycin) and −AB (antibiotic free). The choice of the chelating agents’ concentration was based on their previously reported antimicrobial activity [19].

### 4.2. Determination of Microbial Resistance to the Extender

An *Enterococcus faecalis* strain, previously isolated in our laboratory from rabbit seminal plasma, was used as a sentinel organism. Antimicrobial activity testing of the extender was conducted using plastic (virgin polystyrene) microtiter plates containing the extenders. After subculture from a frozen stock culture stored at −80 °C, each well of the plate was inoculated with 10^3^ and 10^5^ CFU/mL. The test was performed in triplicate. The endpoint was determined after incubation at 37 °C for 24 h. The evaluation was performed manually by assessing turbidity at the bottom of the well indicating growth, or as a clear well indicating growth inhibition.

### 4.3. Enterococcus faecalis Counts in the Extenders After Refrigeration

Using *Enterococcus faecalis* as a sentinel organism, bacterial growth in artificially infected extenders was valuated through bacterial enumeration. A bacterial inoculum (10^3^ CFU/mL) was added to each extender, and the samples were refrigerated at 16 ± 1 °C for 72 h (Dometic Osaka OK15, Humeco, Huesca, Spain). At 24, 48, and 72 h of refrigeration; plate counts were performed using a tenfold dilution series in duplicate onto non-specific Columbia CNA agar with 5% Sheep Blood, Improved II (BD, Becton Dickinson, Madrid, Spain). The plates were then incubated at 37 ± 1 °C for 24 h, after which colonies with typical morphology were counted. The least diluted pair of plates with an average of 30 to 150 colonies was used to calculate the bacterial concentration (CFU/mL). A total of four biological replicates were conducted.

### 4.4. Experiment 1: In Vitro Evaluation

#### 4.4.1. Animals

Twelve New Zealand White male rabbits were housed individually in flat-deck cages, maintained under a controlled photoperiod of 16 h of light and 8 h of darkness. The animals were provided with a standard diet containing 17.5% crude protein, 2.3% ether extract, 16.8% crude fiber, and 2600 Kcal DE/kg, with unrestricted access to water.

#### 4.4.2. Semen Preparation

One ejaculate per male was collected using an artificial vagina. Strict attention was paid to the collection equipment and semen samples were collected into sterile tubes. A subjective evaluation of sperm quality was conducted to determine the initial seminal characteristics. Only ejaculates with a white coloration, motility rates exceeding 70%, intact acrosomes in at least 85% of the sperm, and less than 15% morphological abnormalities were selected for pooling. Sperm concentration was determined by fixing the sample with a glutaraldehyde solution and examining it under phase-contrast microscopy at 400× magnification using a Thoma-Zeiss counting chamber (Marienfeld, Lauda-Königshofen, Germany). The selected samples were then divided into five equal fractions and diluted with the respective extenders at a 1:5 (*v*:*v*) ratio.

#### 4.4.3. Sperm Assessment

Total sperm motility was assessed using Computer-Assisted Sperm Analysis (CASA) following previously described protocols [51]. The ISAS system (Proiser, Valencia, Spain) was utilized, operating at 10× magnification on the heated stage of a phase-contrast microscope (Nikon, ECLIPSE E200, Izasa Scientific, Barcelona, Spain). A Mackler chamber was loaded with 10 µL of the sample, and five fields containing a minimum of 400 spermatozoa were analyzed. Viability was assessed by evaluating plasma membrane integrity using the LIVE/DEAD sperm viability kit (Molecular Probes^TM^, Eugene, OR, USA). This kit utilizes two DNA-binding fluorescent dyes: a membrane-permeant dye, SYBR-14, and a conventional dead-cell dye, propidium iodide (PI). A working solution was prepared by mixing 1.5 µL of SYBR-14 with 49 µL of TCG. Subsequently, 6 µL of the working solution were combined with 100 µL of the sample and incubated for 10 min at 38 °C. After incubation, 2 µL of PI stain were added, and the sample was analyzed after a 10-min incubation period (38 °C).The stained samples were analyzed using fluorescence microscopy (Zeiss Axioscope 5, Pascual y Furio, Valencia, Spain), classifying 100 spermatozoa as either live (SYBR-14-positive, green) or dead (PI-positive, red).

### 4.5. Experiment 2: In Vivo Evaluation

#### 4.5.1. Animals

Alongside the males previously described, 63 New Zealand White origin females were used in this experiment. Females were housed at the experimental farm of the Universitat Politècnica de València.

#### 4.5.2. Artificial Insemination and Litter Size Traits

Females were inseminated using seminal doses prepared with the four experimental extenders, adjusted to a concentration of 40 million/mL. Estrus synchronization was performed using eCG (12.5 IU, SERIGAN; Vetia Animal Health, Madrid, Spain) treatment administered intramuscularly 48 h before insemination. Ovulation was induced with 1 µg of buserelin acetate (Suprefact; Sanofi-Aventis S.A., Barcelona, Spain) given intramuscularly.

Pregnancy rate at day 12 and prolificacy (total number of kits born) were evaluated in inseminated does. Pregnancy rate was diagnosed by abdominal palpation, and embryo and fetal survival rates were assessed by laparoscopy in 7 pregnant does per group. Twelve days post-AI, animals were anesthetized via an intramuscular injection of xylazine (4 mg/kg; Bayer AG, Leverkusen, Germany), followed 5–10 min later by an intravenous injection of ketamine hydrochloride (0.4 mL/kg; Imalgène 500, Merial SA, Lyon, France) into the marginal ear vein. During the laparoscopy procedure, morphine hydrochloride (3 mg/kg; Morfina, B. Braun, Barcelona, Spain) was administered intramuscularly for analgesia. Post-laparoscopy, the animals received antibiotic treatment with gentamicin (4 mg/kg every 24 h for 3 days; 10% Ganadexil, Invesa, Barcelona, Spain) and analgesics, including buprenorphine hydrochloride (0.03 mg/kg every 12 h for 3 days; Buprex, Esteve, Barcelona, Spain) and meloxicam (0.2 mg/kg every 24 h for 3 days; Metacam 5 mg/mL, Norvet, Barcelona, Spain). The number of *corpora lutea* (CL) and implanted embryos (IE) at 12 d and total kits born (TB) were recorded per doe. Embryonic and fetal losses were calculated, respectively, as the proportion of potential embryos produced (number corpora lutea) that did not reach the implantation or fetal stage in all ovulating does [(CL-IE)/CL × 100, (CL-TB)/CL × 100]. Fetal losses were defined as the proportion of implanted embryos that did not reach the fetal stage in pregnant does [(IE-TB)/IE × 100]. Pregnancy diagnosis was performed by abdominal prate (number of kindling’s/numbers of inseminated does) and prolificacy (total number of kits born) was evaluated.

### 4.6. Statistical Analysis

The bacterial counts determination and sperm measurements (motility and viability) were analysed by a generalized linear model including as fixed effects the extender (EDTA, Chitosan, EDTA and Chitosan, +AB and −AB) and time point of refrigeration (0, 24, 48 and 72 h). The analysis was performed with SPSS 27.0 software package (SPSS Inc., Chicago, IL, USA, 2002).

The relevance of the differences between extenders for proportional data (pregnancy rate, and embryonic and fetal losses) or continuous data (prolificacy) was estimated using Bayesian inference. Binomial data for fertility, implantation rate, total kits born, and embryonic and fetal losses were assigned a value of 1 if positive development was achieved, or 0 if it was not. Bayesian inference is based on probabilities, providing great flexibility to construct all kinds of confidence intervals with a chosen probability. In all cases, the progeny origin was included as an extender with five levels (EDTA, Chitosan, EDTA and Chitosan, +AB and −AB). Bounded flat priors were assigned to all unknown parameters, and marginal posterior distributions were estimated using Gibbs sampling. Following preliminary analyses, the final results were derived from Markov Chain Monte Carlo (MCMC) simulations comprising 60,000 iterations, with a burn-in period of 10,000 and a thinning interval of 10. Summary statistics of the marginal posterior distributions were directly computed from the retained samples. Convergence was assessed using the Geweke Z-score criterion, and Monte Carlo sampling errors were evaluated via time-series methods. In all cases, Monte Carlo Standard Errors were minimal, and the Geweke test did not indicate any issues with convergence. The statistics obtained from the marginal posterior distributions of the phenotypic differences between experimental groups were the mean of the difference D_(+AB)-(extenders)_; computed as antibiotic extenders–alternative extenders without antibiotics), the probability of the difference being greater than 0 when D_(+AB)-(extenders)_ > 0 or lower than 0 when D_(+AB)-(extenders)_ < 0 (P0), and the highest posterior density interval at 95% of probability (HPD95%). (+AB)-(extender) estimated the mean of the differences between extender with antibiotics and other extenders traits, P0 estimated the probability of (+AB)-(extenders) ≠ 0, and HPD95% estimated the accuracy. Statistical differences were considered if |(+AB)-(extender)| surpassed the relevant value (R; proposed as one-third of the SD of the trait) and P0 > 0.8 (80%). A Bayesian approximation of the false discovery rate (FDR) was employed, leveraging the cumulative posterior error probability (PEP), analogous to the q-value, to define thresholds for identifying relevant taxa [52]. The PEP was computed as (1 − P0)/0.5, with a cumulative PEP threshold set at 0.05. This threshold implies that approximately 5% of the variables identified as significant were expected to be false positives [52]. Statistical analyses were conducted using the Rabbit R package 1.0 (https://github.com/marinamartinezalvaro/RabbitR, accessed on 1 December 2024), developed by the Institute for Animal Science and Technology (Valencia, Spain). Further details regarding these methodologies are available in a previous review [53].

## 5. Conclusions

The addition of Chitosan (0.05%) to the extender significantly reduces the *Enterococcus faecalis* load at comparable levels of samples treated with penicillin (100 UI/mL) and streptomycin (100 µg/mL). In addition, Chitosan’s protective effect on sperm function, makes it a promising alternative antimicrobial agent for commercial AI in rabbits. Although we have demonstrated its antibacterial activity against *Enterococcus faecalis* (Gram-positive), and previously against *Enterobacteriaceae* (Gram-negative), the next steps should be directed towards determining its spectrum against the bacteria and fungi present in rabbit seminal plasma and its effects on the health and microbiome of the female reproductive tract of females inseminated with diluents containing Chitosan.

## Figures and Tables

**Figure 1 antibiotics-14-00055-f001:**
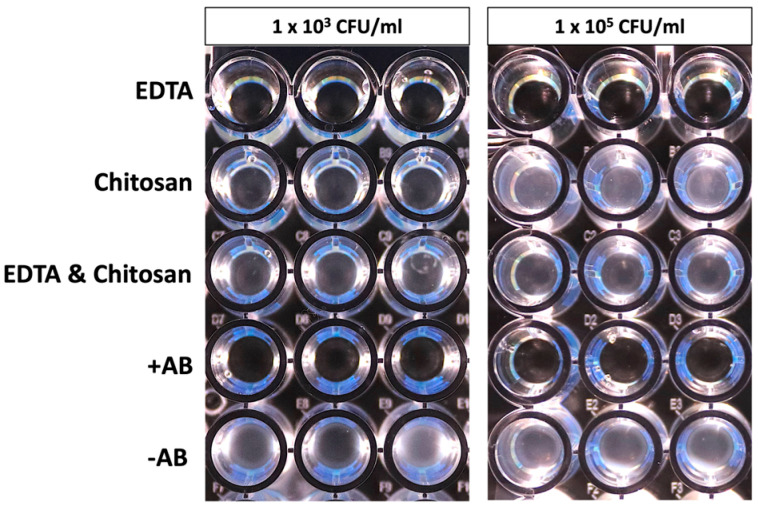
Undetected (translucent well) and detected (turbid well) growth of *Enterococcus faecalis* observed through manual inspection in wells containing rabbit semen extenders after 24 h of incubation at 37 °C. EDTA: TCG (250 mM tris(hydroxymethyl)aminomethane, 83 mM citric acid, 50 mM glucose, pH 6.8–7.0) supplemented with 20 mM ethylenediaminetetraacetic acid. Chitosan: TCG supplemented with 0.05% Chitosan. EDTA and Chitosan: TCG supplemented with 20 mM ethylenediaminetetraacetic acid and 0.05% Chitosan. +AB: TCG supplemented with 100 mg/mL penicillin and 100 mg/mL streptomycin. −AB: TCG without antibiotics.

**Figure 2 antibiotics-14-00055-f002:**
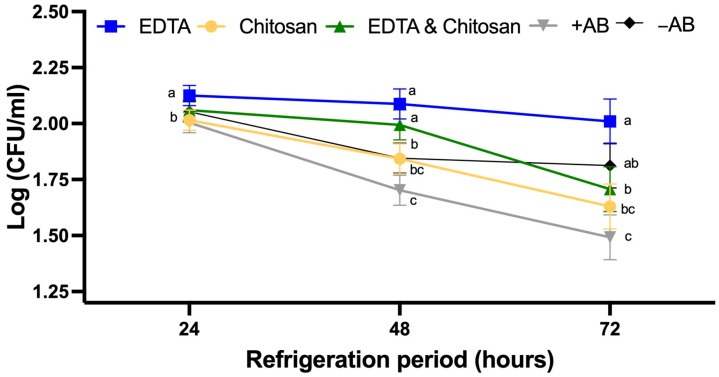
CFU count of various antimicrobial substances in a medium composed of Tris-Citric-Glucose (TCG) against *Enterococcus faecalis* at 24, 48, and 72 h of incubation. Data are expressed as mean ± standard error of the mean. Ethylenediaminetetraacetic acid (EDTA): TCG supplemented with 20 mM ethylenediaminetetraacetic acid. Chitosan: TCG supplemented with 0.05% Chitosan. EDTA and Chitosan: TCG supplemented with 20 mM ethylenediaminetetraacetic acid and 0.05% Chitosan. +AB: TCG supplemented with 100 mg/mL penicillin + 100 mg/mL streptomycin. −AB: TCG without antibiotics. TCG: 250 mM tris(hydroxymethyl)aminomethane, 83 mM citric acid, 50 mM glucose, pH 6.8–7.0. A total of 4 replicates were performed. Different letters indicate statistically significant differences within the same time point (*p* < 0.05).

**Figure 3 antibiotics-14-00055-f003:**
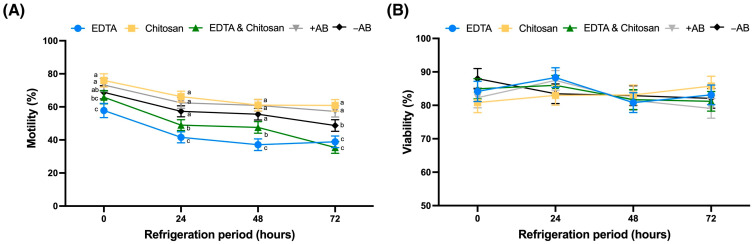
Rabbit sperm characteristics (motility (**A**) and viability (**B**)) in extenders at 24, 48, and 72 h of refrigeration (least square means ± standard error of the mean). EDTA: TCG supplemented with 20 mM ethylenediaminetetraacetic acid. Chitosan: TCG supplemented with 0.05% Chitosan. EDTA and Chitosan: TCG supplemented with 20 mM ethylenediaminetetraacetic acid and 0.05% Chitosan. +AB: TCG supplemented with 100 mg/mL penicillin + 100 mg/mL streptomycin. −AB: TCG without antibiotics. TCG: 250 mM tris(hydroxymethyl)aminomethane, 83 mM citric acid, 50 mM glucose, pH 6.8–7.0. A total of 8 replicates were performed. Different letters indicate statistically significant differences within the same time point (*p* < 0.05).

**Table 1 antibiotics-14-00055-t001:** Effect of antibiotics and chelating agents on reproductive outcomes (pregnancy rate, prolificacy, and embryonic and fetal losses) in rabbit artificial insemination doses (mean ± standard deviation).

	Pregnancy Rate (%)	Prolificacy	Embryonic Losses (%)	Fetal Losses (%)
EDTA	57.52 ± 11.41	9.14 ± 1.25	3.41 ± 3.79	28.73 ± 5.25
Chitosan	76.81 ± 11.73	8.80 ± 1.09	9.50 ± 3.66	27.59 ± 5.22
EDTA and Chitosan	86.81 ± 10.74	5.24 ± 0.95	46.53 ± 3.78	54.21 ± 6.70
+AB	84.65 ± 11.99	9.19 ± 1.01	15.22 ± 3.89	31.30 ± 5.79
−AB	87.34 ± 15.09	8.96 ± 1.29	9.20 ± 4.77	18.49 ± 6.44

**Table 2 antibiotics-14-00055-t002:** Results of the differential abundance analysis comparing the addition of antibiotics and chelating agents to extenders for rabbit seminal AI doses, in relation to pregnancy rate and prolificacy, using Bayesian statistical models. The columns are the posterior mean of the differences among the two groups of the comparison, the probability of the difference being larger (if the difference is positive) or smaller (if negative) than 0 (P0), the highest posterior interval density of 95% (HPD95), and the cumulative posterior error probability (PEP).

Extenders Comparison	Pregnancy Rate				Prolificacy			
	Mean Difference	P0	HPD95%	PEP	Mean Difference	P0	HPD95%	PEP
+AB vs. EDTA	0.27	0.94	[−0.05;0.59]	0.10	0.05	0.51	[−2.82;3.52]	0.97
+AB vs. Chitosan	0.08	0.68	[−0.27;0.38]	0.63	0.38	0.61	[−2.46;3.42]	0.20
+AB vs. EDTA and Chitosan	−0.02	0.55	[−0.32;0.31]	0.88	3.93	0.99	[1.11;6.62]	0.00
+AB vs. −AB	0.02	0.55	[−0.36;0.40]	0.89	0.25	0.56	[−2.91;3.48]	0.29

+AB: TCG (250 mM tris(hydroxymethyl)aminomethane, 83 mM citric acid, 50 mM glucose, pH 6.8–7.0) supplemented with 100 mg/mL penicillin and 100 mg/mL streptomycin. EDTA: TCG supplemented with 20 mM ethylenediaminetetraacetic acid. Chitosan: TCG supplemented with 0.05% Chitosan. EDTA and Chitosan: TCG supplemented with 20 mM ethylenediaminetetraacetic acid and 0.05% Chitosan. −AB: TCG without antibiotics.

**Table 3 antibiotics-14-00055-t003:** Results of the differential abundance analysis comparing the addition of antibiotics and chelating agents to extenders for rabbit seminal AI doses, in relation to embryonic and fetal losses, using Bayesian statistical models. The columns are the posterior means of the differences among the two groups of the comparison, the probability of the difference being larger (if the difference is positive) or smaller (if negative) than 0 (P0), the highest posterior interval density of 95% (HPD95), and the cumulative posterior error probability (PEP).

Extenders Comparison	Embryonic Losses (%)				Fetal Losses (%)			
	Mean Difference	P0	HPD95%	PEP	Mean Difference	P0	HPD95%	PEP
+AB vs. EDTA	0.11	0.98	[0.01;0.22]	0.02	0.02	0.63	[−0.12;0.19]	0.74
+AB vs. Chitosan	0.05	0.83	[−0.05;0.15]	0.34	0.04	0.69	[−0.12;0.18]	0.61
+AB vs. EDTA and Chitosan	−0.31	1.00	[−0.41;−0.20]	0.00	−0.23	0.99	[−0.40;−0.06]	0.00
+AB vs. −AB	0.06	0.85	[−0.05;0.15]	0.29	0.13	0.93	[−0.05;0.28]	0.12

+AB: TCG (250 mM tris(hydroxymethyl)aminomethane, 83 mM citric acid, 50 mM glucose, pH 6.8–7.0) supplemented with 100 mg/mL penicillin and 100 mg/mL streptomycin. EDTA: TCG supplemented with 20 mM ethylenediaminetetraacetic acid. Chitosan: TCG supplemented with 0.05% Chitosan. EDTA and Chitosan: TCG supplemented with 20 mM ethylenediaminetetraacetic acid and 0.05% Chitosan. −AB: TCG without antibiotics.

## Data Availability

The original contributions presented in the study are included in the article, and further inquiries can be directed to the corresponding authors.

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
