# Peer review of "Chitosan-Based Semen Extenders: An Approach to Antibiotic-Free Artificial Insemination in Rabbit"

_antibiotics, 2025, doi:10.3390/antibiotics14010055_

Round 1
Reviewer 1 Report
Comments and Suggestions for Authors
The current study is a well-design experimental approach to test the efficacy of diverse antibiotics-replacement in the rabbit farming, by using EDTA and chitosan-bases extenders.
This robust approach includes not only the study of sperm parameters but also the efficacy of such treatments in the fertility performance. Overall, the chitosan treatment appears to be a potential replacement of antiobiotics, depite not being to inhibit bacterial growth (E. faecalis) at 37 °C for 24 h.
I have some suggestions that need to be addressed by authors to clarify, in my opinion, the manuscript content:
Did you consider including more analysis of bacterial growth apart from “E. faecalis”?
In Figure 3 A, there is a significative effect on sperm motility immediately after dilution with EDTA, EDTA & Chitosan? How can you explain this issue? Could this be due to the concentrations used in your experimental approach?
In Figure 3, despite not being analyzed in your statistical procedure, as long as I understand from the information in the manuscript, how do you explain a “plausible” increase in viability through the incubation period? E.g. Chitosan at 0 h looks lower than 72 h incubation. It is also evident this controversial finding in the EDTA group where it looks like viability increase from 0 to 24 h.
Minor issues:
LIN 304. “Mackler”.
LIN311-312. Did you analyze the sample immediately without incubating PI with the sample? In addition, did you check if the 1 % glutaraldehyde interferes with the staining SYBR-14 and/or PI?
Reviewer 2 Report
Comments and Suggestions for Authors
Marco-Jiménez et al., analyzed in their study the potential use of EDTA and Chitosan on reproductive performance of does inseminated with liquid rabbit semen in alignment with the standards of rabbit semen production centers. The authors of this study contributed to the growth of knowledge in the area of ​​the influence of chitosan on the properties of rabbit sperm and reproductive performance of female rabbits. The researchers showed that the mentioned substance not only has a significant positive effect on the motility of sperm preserved in a liquid state, but also significantly increases the fertility and fecundity parameters of rabbits inseminated in field conditions. They also demonstrated that Chitosan has significantly higher chelating efficacy, and therefore also better protective efficacy for rabbit sperm and embryos, compared to DMSO.
I think that the statistics have been developed well, but I miss tables that would clearly and simply present the reproductive performance indicators of female rabbits.
It seems to me that the authors confused EDTA with Chitosan in the results (lines 93-97) and discussion (lines 188-195)?
